# Influence of Earlier Snowmelt on the Seedling Growth of Six Subboreal Tree Species in the Spring

Erica Marumo [1], Miki U. Ueda [2], Osamu Seki [3], Kentaro Takagi [4] and Kobayashi Makoto [4,*,†]

1 Graduate School of Environmental Science, Hokkaido University, Tokuda 250,
Nayoro 096-0013, Hokkaido, Japan
2 Department of Chemical Biological Sciences, Faculty of Science, Japan Women's University, Mejirodai 2-8-1,
Bunkyo-ku, Tokyo 112-8681, Japan
3 Institute of Low Temperature Science, Hokkaido University, Kita 19 Nishi 8,
Sapporo 060-0819, Hokkaido, Japan
4 Teshio Experimental Forest, Hokkaido University, Toikanbetsu 135, Horonobe 098-2943, Hokkaido, Japan
* Correspondence: makoto@fsc.hokudai.ac.jp; Tel.: +81-1654-2-4264
† Current address: Nayoro Forest Research Office, Hokkaido University, Tokuda 250,
Nayoro 096-0013, Hokkaido, Japan.

**Abstract:** Climate warming is advancing snowmelt timing in the spring at high latitudes. To predict tree growth in subboreal forests under warmer climates based on mechanistic understanding, it is important to assess how advancing snowmelt influences tree growth in the spring via ecophysiological changes in subboreal forests. In this study, we conducted a field manipulation experiment of snowmelt timing and investigated the response of tree growth, leaf functional traits, and bud-burst phenology in the spring for the seedlings of six dominant tree species in subboreal forests. We found that the spring growth of only one species (*Kalopanax septemlobus*) out of six species responded positively to advancing snowmelt. Among the leaf functional traits (leaf mass per area, leaf nitrogen content, leaf $\delta^{13}$C value, leaf dry matter content, and leaf area) and bud-burst phenology, only the increase in leaf area was linked to the enhanced shoot growth of *K. septemlobus*. The significant change in *K. septemlobus* might be associated with its ecological characteristics to prefer regeneration in canopy gaps. These results indicate that advancing snowmelt under warmer winters can be beneficial for tree species that can plastically develop leaf area in Japanese subboreal forests.

**Keywords:** winter climate change; plant and soil interaction; plasticity; functional traits; reforestation

## 1. Introduction

The global climate is becoming warmer [1], with greater changes in winter than in summer in high-latitude ecosystems, including boreal forests [2]. In the past two decades, the northern high-latitude (>40° N latitude) land surface winter climate has experienced some of the most rapid changes on Earth, warming by almost 2.5 times the global average temperature change [3]. As a consequence, spring snowmelt timing had advanced by two weeks in 2010 compared to 1930 in subboreal forests of northern Japan [4,5] and is projected to be even earlier in the future [6]. The snowmelt timing in the spring significantly influences tree growth via eco-physiological mechanisms [7]. In particular, the growth and eco-physiological activity of small trees (seedlings) buried under snowpack is largely determined by snowmelt timing [8,9]. To predict the growth of tree seedlings, which are important individuals of the future tree community, based on mechanistic understanding under a warmer climate, it is important to assess how advancing snowmelt by climate warming influences the spring growth of tree seedlings via ecophysiological changes in subboreal forests.

Advancing snowmelt influences resource availability and environmental stress for tree seedlings during the spring in several ways [10]. In general, earlier snowmelt (1) prolongs

the exposure of seedlings to sunlight [5], (2) accelerates soil nitrogen (N) mineralization via earlier warming of soil or even via increased freeze–thaw cycles [5,11], (3) prolongs the exposure of seedlings to cold air by the absence of snow cover [12,13], and (4) decreases soil moisture [14] in the spring. These changes in the abiotic environment by advancing snowmelt drive differential influences on seedling growth via eco-physiological changes, such as the responses of leaf functional traits and leaf phenology in the spring [15–18]. In other words, leaf functional traits and leaf phenology can be a cue to understand the underlying mechanism of the influence of advancing snowmelt on spring seedling growth [19,20]. Field manipulation experiments have been used to investigate the causal influence of advancing snowmelt on plant traits, leaf phenology, and growth. In a subboreal forest of Japan, advancing snowmelt increases leaf N content via an increase in N uptake, which results in shoot elongation of dwarf bamboo [5]. In a German temperate forest, the snow reduction link to the increase in N uptake was not by the mature trees but by the saplings of a beech species [21]. In a German alpine grassland, advancing snowmelt increased leaf mass per area (LMA) and leaf N content for 11 graminoids and 6 legume species via increased light and N availability [22]. For a perennial herb species in a US alpine ecosystem, advancing snowmelt decreased LMA [23]. For a subalpine herb species in the US, earlier snowmelt decreased the leaf water content and photosynthetic rate via a decrease in soil moisture [24]. Furthermore, although it was an observational study, half of the 101 tested temperate tree species advanced the bud-burst timing by earlier snowmelt due to an extreme warm event during winter in a botanical garden in the US temperate region [18]. However, these studies tested only a part of the three factors (functional traits, leaf phenology, or growth) mainly for herbaceous species, and the relationship between these three factors for tree species is not well explored.

Furthermore, advancing snowmelt can influence seedlings differently among species. For instance, while the growth of the shrub species *Vaccinium myrtillus* in arctic tundra increases with earlier snowmelt [25], that of *Betula nana* and *Salix pulchra* does not change with snowmelt change [26]. In a subboreal forest, earlier snowmelt enhanced dwarf bamboo growth, while birch did not change its growth [5]. The driver of differential growth response to advancing snowmelt could be either (1) different among the species owing to the interspecific variation of stress tolerance and resource use strategy or (2) consistent due to the deterministic change of the environment for tree species by advancing snowmelt. However, the interspecific variation in the plastic responses of spring growth to advancing snowmelt and its underlying eco-physiological mechanism has been little explored among tree species [18].

In this study, we conducted a field manipulation experiment of snowmelt timing and investigated the response of seedling growth of six dominant tree species in subboreal forests in northern Japan. By considering the multiple positive (e.g., longer exposure of seedlings to sunlight and acceleration of soil N mineralization) and negative (e.g., longer exposure of seedlings to cold air and decreases in soil moisture) pathways through which advancing snowmelt influences trees, we hypothesized that (1) the effect of advancing snowmelt on seedling growth, phenology, and leaf functional traits is species-specific among tree species and (2) the species-specific plastic response of seedling growth to advancing snowmelt is related to changes in leaf functional traits and leaf phenology. To test these hypotheses, we studied seedling performance (shoot growth, phenology, LMA, leaf N content, leaf dry matter content, leaf $\delta^{13}$C value, and leaf area) in the spring because spring growth is key for annual plant performance in the target ecosystem [27,28] and it is vulnerable under winter climate change [18].

## 2. Materials and Methods

### 2.1. Study Site

We conducted the manipulation experiment of snowmelt timing in the nursery of the Teshio Experimental Forest of Hokkaido University in northern Japan (44°55′06′′ N, 142°01′20′′ E, 15 m a.s.l.). The mean annual temperature and minimum air temperature

were approximately 5.9 °C and −23.0 °C, respectively (between 2016 and 2017). The annual precipitation in 2016 and 2017 was 1004 mm and 871 mm, respectively (Toyotomi meteorological station). Thirty percent of the annual precipitation falls as snow from November to April. Snow covered the forest floor from November to early April. Just before the experiment, the snow depth was 90 cm in the nursery. The soils of this region rarely freeze, despite cold air temperatures, because the deep snow insulates the soil from air at sub-zero temperatures throughout winter [29].

### 2.2. Plant Cultivation

Six plots (each plot was 5 m by 5 m) were established in the nursery. We purchased tree seedlings from a commercial seedling company (Snow brand seed Co., Ltd., Sapporo, Japan), and tree seedlings shorter than 50 cm in height were planted in May 2016. Ditches (50 cm wide and 50 cm deep) were dug between the plots to prevent the tree roots from accessing the other plots. The planted species were *Acer mono* Maxim., *Betula platyphylla* Skatchev, *Cornus controversa* Hemsl, *Kalopanax septemlobus* Koidz, *Quercus crispula* Blume, and *Ulmus davidiana var. japonica* (a total of six species) which are all dominant broad-leaved species in northern Hokkaido. Ten individuals of each species were randomly planted within each plot (a total of 60 individuals for each species). The distance between the seedlings was 50 cm so that the canopies did not influence each other. After waiting for one year after plantation, the sampling and measurements were conducted in the spring of 2017.

### 2.3. Snowmelt Treatment

At the end of winter when the snow depth reached its maximum (end of March), the snowmelt treatment was conducted for three out of six plots (hereafter SM plots) (please see the details of the method in [5]). The other three plots were set as control plots (Con plots) without any snowmelt treatment. To melt the snow, large heaters (HGDHII, HOTGUN, Shizuoka Seiki Co., Fukuroi, Japan) were used. A 20 m-long duct was attached to each heater and extended upon the surface of snowpack across an SM plot by folding the duct to fit within a 5 m by 5 m area. From the small holes throughout the duct, warm air (30–40 °C) was supplied to the surface of the snowpack to induce snowmelt. The duct was covered with a polyethylene sheet to keep the warm air within the plot and to melt the snow there efficiently. Heating was conducted over one day until the snow depth decreased from 90 cm to 30 cm. Because the tree seedlings were bent under the snow [8], the warm air did not hit the seedlings directly. The timing of snowmelt was determined as the date when half of the area of snow cover in each plot disappeared.

### 2.4. Environmental Measurements

Air temperature, soil temperature, soil moisture, and soil inorganic N contents were monitored as abiotic environmental factors. The air temperature was measured every 1 h at the climate station within the nursery from March to June 2017. For the soil temperature and moisture, every 1 h, we started to measure soon after the snowmelt timing in snowmelt plots at 10 cm in soil depth by using digital TDT moisture and temperature sensors (SDI-12, Acclima, Meridian, ID, USA). By using the hourly data, the daily average was calculated for each day. The soil moisture at 10 cm in depth is known to change sensitively to snowmelt [14], and roots are abundantly distributed in our region [30].

For soil inorganic N (the sum of $NH_4^+$ and $NO_3^-$N), three subsamples were obtained from each plot to 10 cm in depth to make a composite in each plot. This composite of soil was sieved with a 4.75 mm sieve to remove the roots and stones from the soil and homogenize the soil. The sieved soil (8 g) was shaken with 40 mL of 2 M KCl over 1 h and filtered with filter paper. The contents of $NH_4^+$ and $NO_3^-$N in the filtered solution were measured with an autoanalyzer (AACS-4, BL-TEC Co., Ltd., Osaka, Japan). The contents of inorganic N in a certain amount of dry soil were calculated based on the soil moisture content.

### 2.5. Plant Growth

By avoiding the individuals that were planted at the edge of each plot, three individuals were selected for the measurements for each species in each plot. For each individual, three current year's shoots were measured in the upper part of the canopy. For the case of *K. septemlobus*, because only one of the branches was a long shoot and the others were short shoots within the canopy of the measured seedlings, only one current-year long shoot was measured. As a proxy for spring growth, we measured the annual shoot length in late June.

### 2.6. Phenology

From April to June 2017, we measured the bud-burst timing every 2–7 days depending on the season for all planted individuals. The date of bud-burst was identified when the buds across half of the canopy opened [31].

### 2.7. Leaf Traits

Three leaves were obtained from each individual whose shoot growth was measured. For some individuals with fewer leaves, only one leaf was collected from each individual to avoid significant destructive damage to the growth of seedlings. For leaf traits, we calculated the parameters as below.

$$\text{Leaf mass per area (LMA)} = \text{leaf dry mass/leaf area (g/m}^2)$$

$$\text{Leaf dry matter content (LDMC)} = \text{leaf dry weight/leaf fresh weight (unitless)}$$

$$\text{N}_{area} = \text{leaf N content/leaf area (g/m}^2)$$

The fresh leaves were weighed at 0.1 mg resolution with a balance, scanned at 300 dpi resolution with a scanner (GT-S630, Epson, Suwa, Japan), and analyzed with the free image software ImageJ 1.6.0. The leaf dry weight was measured after drying the leaf at 70 °C for three days. By dividing the leaf area data by the dry weight of the same leaf sample, LMA was calculated. LDMC was calculated with the leaf fresh weight and leaf dry weight data. Then, the dried samples were ground, and their N contents were analyzed with an elemental analyzer (Fisons Instruments, NA1500 NCS, Glasgow, UK) and calculated as an area base. Furthermore, the $\delta^{13}$C value was analyzed with an isotope ratio mass spectrometer (Delta plus, Finnigan MAT, Apt, France). The natural abundance of $^{13}$C (hereafter referred to as $\delta^{13}$C) was calculated as follows:

$$\delta^{13}\text{C} = [\{\text{R}_{sample}/\text{R}_{standard}\} - 1] \times 1000\ ‰,$$

where R is the $^{13}$C/$^{12}$C ratio and the standard was the PDB standard [32]. The summary of the species-specific average value of the functional traits and phenology were shown in Table 1.

**Table 1.** The average value (±SD) of the functional traits of six species tested in our study.

| Species | Wood Structure | Bud-Burst Day (DOY) | Leaf Area (m$^2$) | LMA (g/m$^2$) | LDMC | Leaf N (g/m$^2$) | $\delta^{13}$C (‰) |
|---|---|---|---|---|---|---|---|
| *Acer mono* | Diffuse | 140.2 ± 4.5 | 0.12 ± 0.05 | 33.9 ± 14.4 | 0.37 ± 0.05 | 1.40 ± 0.56 | −27.5 ± 0.2 |
| *Betula platyphylla* | Diffuse | 131.3 ± 2.1 | 0.09 ± 0.02 | 49.9 ± 6.2 | 0.34 ± 0.08 | 2.05 ± 0.22 | −29.6 ± 0.3 |
| *Cornus controversa* | Diffuse | 136.0 ± 3.7 | 0.07 ± 0.05 | 44.0 ± 16.8 | 0.45 ± 0.11 | 1.90 ± 0.29 | −28.1 ± 1.8 |
| *Kalopanax septemlobus* | Ring | 140.1 ± 1.8 | 0.20 ± 0.12 | 56.7 ± 22.0 | 0.34 ± 0.07 | 2.57 ± 0.37 | −25.9 ± 0.4 |
| *Quercus crispula* | Ring | 142.2 ± 3.9 | 0.08 ± 0.04 | 40.1 ± 6.3 | 0.32 ± 0.08 | 1.75 ± 0.23 | −26.0 ± 0.8 |
| *Ulmus davidiana* | Ring | 134.5 ± 3.6 | 0.04 ± 0.01 | 43.5 ± 9.4 | 0.32 ± 0.07 | 2.06 ± 0.05 | −28.5 ± 0.5 |

### 2.8. Statistical Analysis

First, for each species, the effect of snowmelt treatment was analyzed with a generalized linear mixed model (GLMM). The response variable was set as shoot length and leaf functional traits, and the explanatory variable was the snowmelt treatment. The ran-

dom effect was the individual ID to consider the pseudoreplication of samples/measurements within each individual. For the influence of snowmelt treatment on the phenology data of each species, we employed a generalized linear model (GLM) instead of GLMM because one data point of bud-burst timing was obtained for each individual. The functions of GLMM and GLM were set as below.

GLMM: Response variable (shoot length or functional trait) ~

Fixed effect (snowmelt treatment) + Random effect (individual ID)

GLM: Response variable (phenology) ~ Fixed effect (snowmelt treatment)

The functions used for GLMM and GLM analyses were "glmer" and "glm" in R software, respectively. The data distribution of these analyses was assumed to follow a normal distribution (family = Gaussian).

Second, for the species with significant differences in shoots by snowmelt treatment, by using the stepAIC function, we conducted stepwise model selection analysis to check the important predictor of the response of shoot growth to advancing snowmelt. The full model included all leaf traits (LMA, LDMC, leaf N contents, $\delta^{13}$C, and leaf area) and phenology (the bud-burst timing) as explanatory variables. The random effect was set as the individual ID. As an exception, for *K. septemlobus*, we conducted GLM instead of GLMM because we collected only one shoot from each individual due to the simple branching pattern of the species, and it was not necessary to consider the within-individual pseudoreplication for shoots. The functions of GLMM and GLM were set as below.

GLMM: Response variable (shoot length) ~

Fixed effect (LMA + LDMC + leaf N contents + $\delta^{13}$C, leaf area)

+ Random effect (individual ID)

GLM: Response variable (shoot length) ~

Fixed effect (LMA + LDMC + leaf N contents + $\delta^{13}$C, leaf area)

For the selected explanatory variables in the best model, we conducted a Wald test to see the significant usefulness of the selected explanatory variable to explain shoot growth. We used lme4 package for the GLMM analysis and all statistical analyses were conducted by using R software version 4.1.0 (R Development Core Team 2021, Vienna, Austria).

## 3. Results

### 3.1. Abiotic Environment

During the investigation, the average daily air temperature was kept above 0 °C after day of the year (DOY) 92 and started to sometimes be above 5 °C after DOY 96 (Figure 1). Snowmelt occurred on DOY 96 in the Con plots and DOY 106 in the SM plots. The snowmelt treatment resulted in earlier snowmelt and coincidental earlier warming of soil temperature and earlier exposure of soil and seedings to air by 10 days (Figure 2a). The soil moisture contents tended to increase by 10% in the snowmelt treatment. During the investigation (Figure 2b). The snowmelt treatment did not show any significant influence on soil inorganic N contents in the spring (Figure 3).

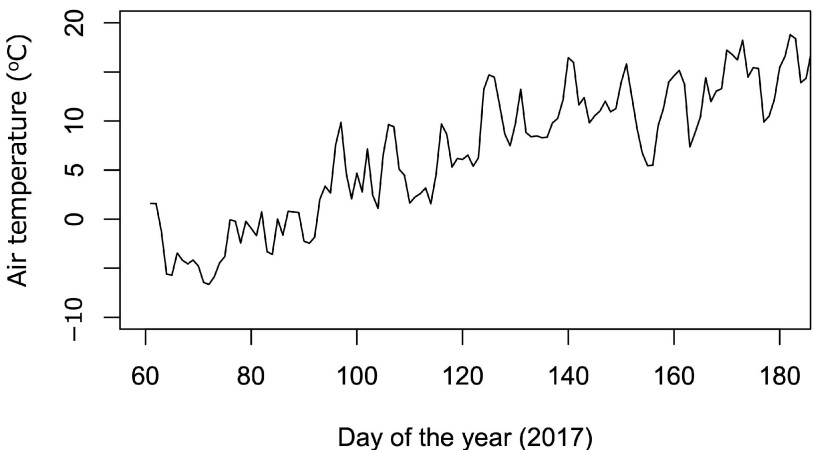

**Figure 1.** The daily average air temperature from March to June in the study site in 2017.

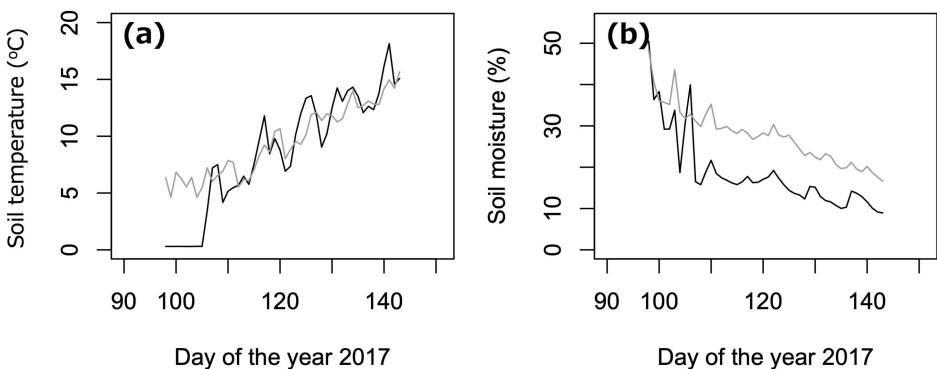

**Figure 2.** The daily average soil temperature (**a**) and moisture (**b**) from April to May 2017. Black line = control (Con) plots and gray line = snowmelt (SM) plots. The data are the average value of two sensors.

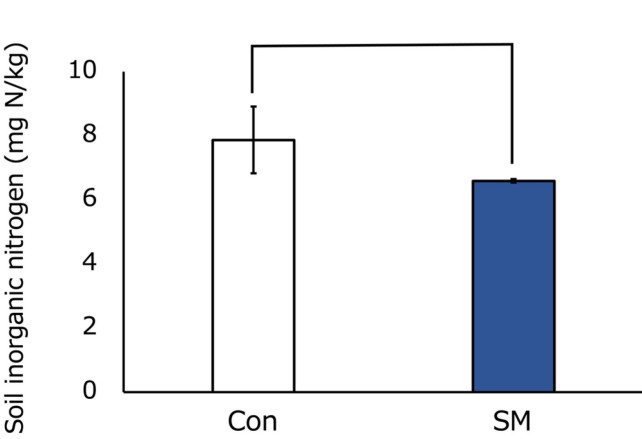

**Figure 3.** The soil inorganic nitrogen contents in the control (Con) and snowmelt (SM) plots in June 2017 (n = 3, error bars are ±SE). The statistical non-significance was tested with a Wald test. n.s. = non-significant.

*3.2. Shoot Growth*

Snowmelt treatment significantly increased the shoot growth of *K. septemlobus* (Table 2, Figure 4 $p < 0.05$). On the other hand, other species did not show any significant change in their shoot growth by snowmelt treatment (Table 2, Figure 4).

**Table 2.** The results of GLMM analysis (for *K. septemlobus*, GLM analysis) (*p*-value and estimated slope) for the effect of snowmelt treatment. Estimate = estimated slope of the models; n.s. = non-significant. The estimates were shown only when the *p*-value (*p*) was less than 0.05.

| Species | Shoot Length | | Bud-Burst Day | | Leaf Area | | LMA | | LDMC | | Leaf N | | Leaf $\delta^{13}$C | |
|---|---|---|---|---|---|---|---|---|---|---|---|---|---|---|
| | Estimate | *p* | Estimate | *p* | Estimate | *p* | Estimate | *p* | Estimate | *p* | Estimate | *p* | Estimate | *p* |
| *Acer mono* | | n.s. | −0.021 | *p* < 0.01 | | n.s. | | n.s. | | n.s. | | n.s. | | n.s. |
| *Betula platyphylla* | | n.s. | −0.021 | *p* < 0.01 | | n.s. | | n.s. | | n.s. | | n.s. | | n.s. |
| *Cornus controversa* | | n.s. | −0.025 | *p* < 0.01 | | n.s. | | n.s. | | n.s. | | n.s. | | n.s. |
| *Kalopanax septemlobus* | 0.57 | *p* < 0.05 | −0.015 | *p* < 0.01 | 0.99 | *p* < 0.01 | −0.26 | *p* < 0.05 | | n.s. | −0.37 | *p* < 0.05 | | n.s. |
| *Quercus crispula* | | n.s. | | n.s. | | n.s. | −0.15 | *p* < 0.05 | | n.s. | | n.s. | | n.s. |
| *Ulmus davidiana* | | n.s. | −0.015 | *p* < 0.01 | | n.s. | | n.s. | | n.s. | | n.s. | | n.s. |

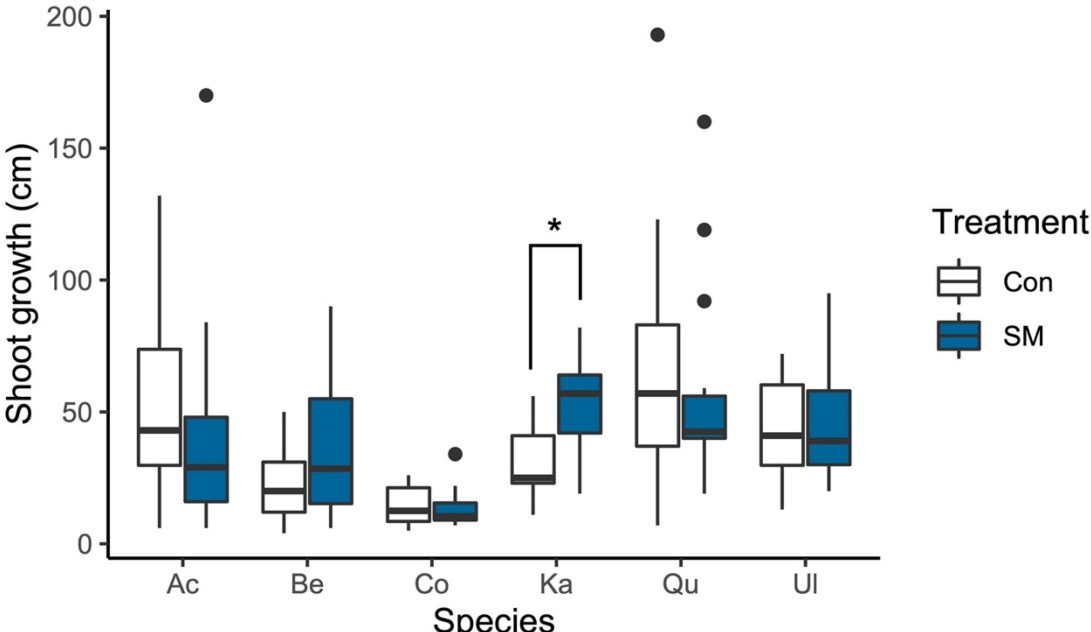

**Figure 4.** The shoot growth of six tree species in June 2017 as influenced by the snowmelt treatment. The white boxes represent the data of the control (Con) and the blue boxes represent the snowmelt (SM) treatment. (n > 9). Ac = *Acer mono*, Be = *Betula platyphylla*, Co = *Cornus controversa*, Ka = *Kalopanax septemlobus*, Qu = *Quercus crispula*, and Ul = *Ulmus davidiana*. *: *p* < 0.05 by GLMM (for *K. septemlobus*, GLM analysis).

*3.3. Phenology and Leaf Traits*

Snowmelt treatment advanced the leaf-out timing of all species (*p* < 0.01, Table 2) except *Q. crispula* (Table 2). Furthermore, the snowmelt treatment increased the leaf area of only *K. septemlobus* (*p* < 0.01, Table 2, Figure 5). LMA increased in *K. septemlobus* and *Q. crispula* with advancing snowmelt (*p* < 0.05, Table 2), while LDMC did not show any significant response to advancing snowmelt among all species (Table 2). The leaf N content significantly decreased in *K. septemlobus* in response to the snowmelt treatment (*p* < 0.05, Table 2).

As the parameter related to shoot growth in the species influenced by snowmelt treatment (in this case, only *K. septemlobus*), leaf area was selected as the significantly important explanatory variable for the shoot response (*p* < 0.05, Table 3). Specifically, a longer shoot was positively associated with leaf area in *K. septemlobus* (Table 3).

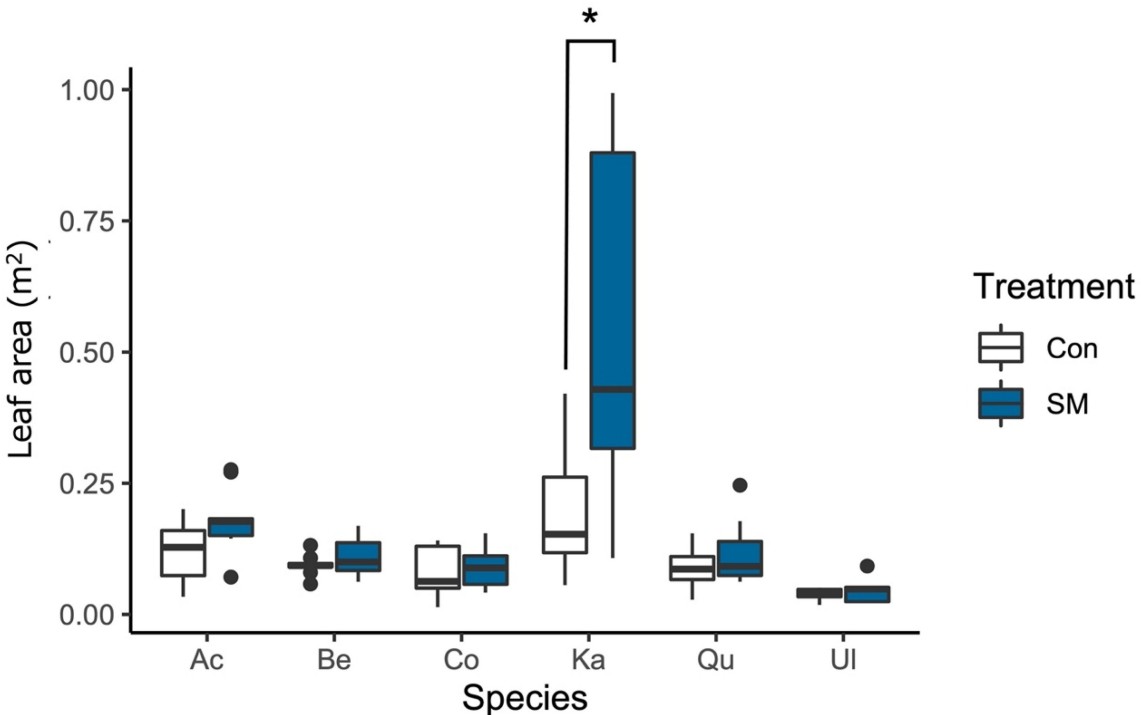

**Figure 5.** The leaf area of six tree species in June 2017 as influenced by the snowmelt treatment. The white boxes represent the data of the control (Con) and the blue boxes represent the snowmelt (SM) treatment. (n = 9). Ac = *Acer mono*, Be = *Betula platyphylla*, Co = *Cornus controversa*, Ka = *Kalopanax septemlobus*, Qu = *Quercus crispula*, and Ul = *Ulmus davidiana*. *: $p < 0.05$ by GLMM (for *K. septemlobus*, GLM analysis).

**Table 3.** The best GLM selected to explain the shoot growth of *K. septemlobus*. n.s. = not significant. The estimated slope and *p*-value were judged with a Wald test.

| Response Variable | Explanatory Variables | | | | | |
|---|---|---|---|---|---|---|
| | **Bud-Burst Day** | **Leaf Area** | **LMA** | **LDMC** | **Leaf N** | **Leaf $\delta^{13}$C** |
| Shoot length | n.s. | Estimate: +46.1 $p < 0.05$ | n.s. | n.s. | n.s. | n.s. |

## 4. Discussion

Consistent with our first hypothesis, the influence of advancing snowmelt on spring growth differed among the six tree species (Figure 5). This result is consistent with a previous study showing the species-specific response of plants to earlier snowmelt in arctic tundra [26], although we assumed that other species also reacted to different degrees. At present, we are not sure why only this species reacts significantly to advancing snowmelt. *K. septemlobus* has the largest leaf area, LMA, and leaf N and the smallest leaf $\delta^{13}$C in the six tested species. Among these functional traits, the species with the large leaf size is known to be adapted to cold climate globally [33]. It is possible that the largest leaf area functioned well in the cold climate in the spring after snowmelt and this resulted in the positive reaction of *K. septemlobus* growth to snowmelt treatment. In future studies, it is necessary to test the relationship between leaf area and early snowmelt by including a greater number of species with a variety of leaf sizes for the analysis. The advancement of snowmelt timing in SM plots could be due to the reduced amount of snow remaining in the SM plots and a shorter time being required to melt as compared to the Con plots.

Consistent with our second hypothesis, we found leaf traits useful to assume the underlying mechanism of growth response to snowmelt treatment. In our study, the advancing snowmelt increased the leaf area of *K. septemlobus* (Figure 5), and the increased

leaf area was linked to the enhancement of shoot growth of *K. septemlobus* (Table 3). A larger leaf area can increase light interception and consequently enhance shoot growth. Because drought stress can generally suppress leaf area [20], the increase in leaf area was attained partly because of the lack of a significant decrease in soil moisture by the snowmelt treatment in our study (Figure 2). Although we expected a decrease in soil moisture by early snowmelt because of the early leaching of snowmelt water into the ground water [14], this was not observed in our study (Figure 2). The snowmelt treatment conducted in the nearby area also resulted in an increasing tendency of soil moisture in northern Hokkaido [5]. Additionally, in eastern Hokkaido, a snow removal experiment also showed an increase in soil moisture [34]. In [34], the soil freeze–thaw cycle due to the absence of snow cover and upward water flux is assumed to be the reason. However, in our study sites, the soil hardly freezes after snowmelt timing because the air temperature becomes warm (minimum air temperature mostly stays above 0 °C) when snow disappears from the forest floor in northern Hokkaido. One possibility is that a part of the water from the melted snow from the snowmelt treatment could refreeze the snow left in the SM plots and the water did not leach into soil. Because the soil moisture condition in spring is crucial for tree growth under a changing climate [35], the underlying mechanisms should be tested. Interestingly, for *K. septemlobus*, leaf N decreased in response to the snowmelt treatment. Because the soil N availability for plants did not change in response to the snowmelt treatment (Figure 3), this decrease might be driven by the dilution effect with enlarging leaves [36,37].

Species other than *K. septemlobus* did not change shoot growth despite the advancement of bud-burst timing (Figure 4). While the earlier start of bud-burst is known to be linked to increased growth by the prolongation of the growing season [12], our results did not align with this. In fact, it has also been reported that earlier bud-burst can result in a higher risk of frost damage in the study region [8]. For the species with no response, it is possible that the benefit from the earlier start of shoot growth was canceled with late frost damage. In fact, the hourly temperature data at the closest meteorological station (Toyotomi station) show that in June, there was a day when the minimum temperature dropped below zero. These results indicate that the advancing bud-burst timing (at least which is caused by the 10 days earlier snowmelt as in this study) is not always linked to the increasing spring growth. These results indicate that the tree seedlings of many of the dominant broad-leaved species in the Hokkaido region are relatively resilient to advancing snowmelt.

We expected that LMA and the leaf $\delta^{13}C$ value would also explain the change in shoot growth by advanced snowmelt in the context of drought stress [17,38]. However, at least for the six species tested in our study, these functional traits were not powerful in explaining the growth response to advancing snowmelt (Table 3). Again, this is partly because of the absence of a reduction in soil moisture by earlier snowmelt.

## 5. Conclusions

In conclusion, advancing snowmelt under warmer winters can be beneficial for certain species that can plastically change their leaf area under changing light conditions for seedlings in the subboreal forests of northern Japan. By considering the fact that seedling growth reacts differently among species to early snowmelt, we should select tree species for plantation by considering whether the selected species can acclimate well or not to warmer winter predicted in near future. As a further study, it is useful to investigate more tree species with a variety of plasticity or light-use strategies to draw a general conclusion about the driver of the species-specific reactions of trees to warmer winter.

**Author Contributions:** Conceptualization, E.M. and K.M.; methodology, O.S. and M.U.U.; formal analysis, K.M.; investigation, E.M.; writing—original draft preparation, E.M.; writing—review and editing, K.M. and K.T.; project administration, K.T.; funding acquisition, K.M. and K.T. All authors have read and agreed to the published version of the manuscript.

**Funding:** This study is supported by Asahi Glass Foundation and a JSPS grant-in-aid for young researchers (15K18708) and a JSPS grant-in-aid for scientific research (21H05316) to M.K.

**Data Availability Statement:** The data presented in this study are available on request from the corresponding author. The data are not publicly available due to the usage for ongoing project.

**Acknowledgments:** We sincerely thank the technical staff of Teshio Experimental Forests for the establishment and management of the nursery.

**Conflicts of Interest:** The authors declare no conflict of interest.

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
