# Peer review of "Influence of Earlier Snowmelt on the Seedling Growth of Six Subboreal Tree Species in the Spring"

_forests, doi:10.3390/f14030600_

Round 1

Author Response

Thanks for your constructive comments. We have revised throughout the manuscript following the comments. Because the reviewer put comments directly on the manuscript, we have shown the reply to each comment also directly on the manuscript. Please refer the attached file.

Reviewer 2 Report

This concise but well written paper reports interesting results found in the experiment conducted regarding 6 tree species in Japan. The study was well designed, and the obtained results were properly analysed. The article is well structured and keeps its logic flow from Introduction to Discussion. Consequently, I do not have any substantial reservations regarding the paper and I recommend its publication in Forests after minor revision.

My comments / suggestions referring to the paper are as follows:

1.       L. 94: Please show the coordinates in more accuracy.

2.       L. 95-96. The sentence needs correction. E.g. this could be “The mean annual temperature and minimum air temperature were approximately…”. The thread of precipitation is picked up in the next sentence.

3.       L. 116: “leached” ?

4.       L. 127 / 125: “half of the snow” does not match “90 cm to 30 cm”

5.       L. 139: “a composite in” -> “a composite sample in”

6.       L. 206: the acronym “DOY” should be described in full when the first time used

7.       L. 207: Did the snowmelt really occur later in control than in SM plots? If it is true, this should be discussed in Discussion (potential reasons and consequences for the experiment).

8.       L. 209 / 210: To facilitate a reader to follow the text I would suggest inserting letters to figures, i.e. Fig. 2a, Fig 2b. In figure 2 “Soil temperature” and “Soil moisture” could be placed just next to “a” and “b” in a figure body.

9.       L. 210: please delate one redundant space

1.   I would prefer if statistical significance would be shown using letters in figure 3, 4 and 5. I stay the decision regarding this issue to journal editors.

1.   Please use more contrast colours for boxes in figs 4 and 5. Green and blue look the same if a reader would dispose only a printed black / white copy.

1.   L. 230: B. platyphylla -> italics

1.   Table 1: a) the table should be referred and described briefly in a text, b) as you show average values, SD or SE values should also be given (e.g. 140±12 etc.).

1.   Table 2: a) “spices” -> species, b) “leaf Narea” “leaf N content” or “leaf N”

1.    Figure 5 should be placed in Results, not in Discussion.

1.   L. 284-285: Please describe in some more detail this interesting and important for your study findings.

1.   I wander about the potential reasons of the higher soil moisture in SM than in control found in your study. This result is interesting and I agree with your reasoning in Discussion regarding this issue. However, we must be aware that after melting the snow partly disappeared in SM, but the pool of water stayed the same (quite similar) in both variants. Probably the water from melted snow could freeze again in the left snow. This (with subsequent consequences) could be consider also in Discussion.

1.   L. 291-292: I agree. Similar “dilution effect” was also found in needles of containerised plantings compared to bare root plantings. Please see Table 2 in Klavina et al. 2013. Survival, Growth and Ectomycorrhizal Community Development of Container and Bare-root Grown Pinus sylvestris and Picea abies Seedlings Outplanted on a Forest Clear-cut. Baltic Forestry 19(1): 39 – 49.

1.   The Conclusions paragraph could be enlarged. It could constitute a separate paper section. You could emphasis also potential importance for silviculture / forest management of your results and directs of further studies in this paper’s section. 

Author Response

We appreciate your comments on our manuscript. Here we upload the list of the comments and replies to all comments. The sentence started with "C" is the reviewer's comment and that with "R" is the reply to the C just above. 
